# Integrating Reinforcement Learning into M/M/1/K Retry Queueing Models for 6G Applications

**DOI:** 10.3390/s25123621

**Published:** 2025-06-09

**Authors:** Djamila Talbi, Zoltan Gal

**Affiliations:** Faculty of Informatics, University of Debrecen, 4032 Debrecen, Hungary

**Keywords:** reinforcement learning, queueing system, 6G, terahertz frequency, deep Q-network, singular value decomposition, high speed networks

## Abstract

The ever-growing demand for sustainable, efficient, and fair allocation in the next generation of wireless network applications is a serious challenge, especially in the context of high-speed communication networks that operate on Terahertz frequencies. This research work presents a novel approach to enhance queue management in 6G networks by integrating reinforcement learning, specifically Deep Q-Networks (DQN). We introduce an intelligent 6G Retrial Queueing System (RQS) that dynamically adjusts to varying traffic conditions, minimizes delays, reduces energy consumption, and guarantees equitable access to network resources. The system’s performance is examined under extensive simulations, taking into account multiple arrival rates, queue sizes, and reward scaling factors. The results show that the integration of RL in the 6G-RQS model successfully enhances queue management while maintaining the high performance of the system, and this is by increasing the number of mobile terminals served, even under different and higher traffic demands. Furthermore, singular value decomposition analysis reveals clusters and structured patterns, indicating the effective learning process and adaptation performed by the agent. Our research findings demonstrate that RL-based queue management is a promising solution for overcoming the challenges that 6G suffers from, particularly in the context of high-speed communication networks.

## 1. Introduction

With the rapid evolution of wireless communication, the 6G era introduces new challenges in resource management. This calls for smarter and more adaptive solutions to ensure efficient, sustainable, and fair allocation of network resources.

### 1.1. Context & Motivation

Sixth-generation (6G) wireless networks are witnessing a rapid evolution that drives an unprecedented demand for certain high-level objective key performance indicators (KPIs), such as efficiency, fair resource allocation, and sustainability. Such a complex network must accommodate a massive growth in connected devices, support ultra-low latency applications, and have the ability to operate under stringent energy constraints.

It is obvious that such aspects increase complexity, which makes traditional queue management techniques insufficient, as they suffer from balance latency, fairness in resource allocation, and energy consumption. A well-created intelligent queueing model will address the challenges mentioned. An intelligent queue refers to adaptive queue management systems that leverage real-time data and advanced algorithms, such as machine learning or reinforcement learning, to dynamically optimize queue handling decisions. These systems can adjust to changing network conditions to improve overall performance beyond static or rule-based methods. Studying how to minimize delays, reduce unnecessary energy consumption, and ensure equitable access to network resources will help in overcoming the existing problems.

Designing such a system will enhance the overall performance and sustainability of next-generation networks. Meanwhile, it requires adaptive decision making based on its dynamic nature, where traffic patterns are unpredictable. Here comes the role of AI-based reinforcement learning (RL). Unlike conventional AI methods, RL allows networks to learn from experience and dynamically optimize the complex system. For the queueing models and complex 6G networks, it helps in managing strategies based on real-time network conditions.

By leveraging RL, intelligent queueing mechanisms can make autonomous, data-driven decisions that continuously enhance over time, leading to a more robust and fair communication system.

### 1.2. Research Gap

Many studies currently focus on packet scheduling and traffic prediction rather than adapting queue management in highly dynamic wireless environment scenarios. Traditional queueing systems often rely on fixed threshold-based loads, where limited attention has been given to the idea of integrating RL within queueing systems, specifically for 6G and THz scenarios, where further complexity is added due to many factors, such as reattempted transmissions, beam misalignment, and intermittent connectivity.

Thus, a significant gap in research appears regarding the role of RL in optimizing the queue management for 6G-based queueing systems under dynamic traffic conditions, as well as the impact of retrial mechanisms on learning-based queueing strategies, and the fairness–energy–latency tradeoff after integrating RL for queue optimization, especially for ultra-reliable and low-latency (URLLC) applications.

### 1.3. Contributions of the Research Study

This research study presents a novel RL-based queue management for 6G application systems, leveraging DQN to help optimize the service decision dynamically. The key contributions of the research study are as follows:Introduction of the 6G retrial queueing system: We develop a code that simulates a simple example of the RQS to address the challenges.Integration of RL in the 6G-RQS: We propose an RL-based decision-making model that dynamically adapts to the nature of the 6G-RQS environment and tries to adapt the service and retrial policies for optimization purposes.Comprehensive performance evaluation: We simulate many different scenarios across varying parameters to see the ability of the model in enhancing the service and the system efficiency.Singular Value Decomposition (SVD) analysis: We employ SVD to extract existing latent patterns in queue behavior that could influence RL decisions.Implications for future 6G and THz networks: We demonstrate how RL manages the proposed model and check how it could contribute to scalable, fair, and energy-efficient scheduling in ultra-dense 6G and for communicating over THz spectrums.

This research lays the groundwork for an AI-driven, self-optimizing network architecture that could adapt to the dynamic nature of next-generation wireless systems.

The structure of this work is as follows: the second section discusses the existing related research studies. The third section introduces the proposed RQS model with an explanation of its main features and presents the simulation results along with the challenges. The fourth section explains the proposed RL model, analyzes the simulation results, and gives a comprehensive discussion of the performance. Finally, the fifth section concludes the overall research findings and proposes potential future directions to build upon this study. 

## 2. Related Work

This section reviews key studies on the presented research and optimization techniques in wireless networks. It lays the foundation for understanding the advancements and limitations that motivate our proposed approach.

### 2.1. Queueing Theory in the Next-Generation Networks

Queueing systems have long been a hot research topic among researchers because of their vast applicability in many fields, particularly in networking and wireless communications, where efficient allocation and congestion control mechanisms are essential. There are many research works that leverage queueing theory to analyze network performance, enhance quality of service, and optimize scheduling policies. A recent study has introduced a queueing theory-based framework for modeling fog computing systems, which helps to optimize the resource allocation to meet the QoS requirements [1]. This model helps to identify the bottlenecks and improve system performance by allowing user-defined parameter adjustments. In the same field, a research group proposed another task allocation method to minimize the latency in mobile edge computing (MEC) for future internet applications with dedicated, shared, and cloud servers [2]. They used queueing theory to model transmission delay and solve an optimization problem, where results show an efficient allocation of tasks and a reduction in the overall delay. However, another study explored cybertwin-based multiaccess edge computing as an alternative to traditional virtual machine (VM) systems, where they show its benefits in cooperating through a control plane, and the results promise future solutions for the upcoming wireless communication network beyond 5G [3].

A comprehensive survey on network delay was conducted in [4] with a focus on the delay, a special key performance indicator of the queueing model, and its impact on communication performance. They examined the delay effect in different networks, including wireless, mobile and IoT and others, offering insights into delay management strategies. Moreover, other research teams have analyzed the performance of different queueing mechanisms in managing network traffic over Wi-Fi. Results highlight the good performance of the weighted fair queueing (WFQ) in allocating bandwidth and reducing packet variation [5]. QoS challenges were explored in [6] in next-generation wireless sensor networks (WSNs) and the 5th generation; the research group proposes a dynamic queueing mechanism for prioritizing traffic based on data type and queue length, and other KPI aspects. The findings show the improvements in QoS efficiency for real-time communication in limited bandwidth networks. Nevertheless, queueing models for cognitive radio networks were examined by focusing on the sensing time of secondary users and their access to idle channels that are originally allocated to primary users [7]. It was found that both models share the same stability condition, with a non-interrupting model having fewer sensing secondary users.

For congestion control in network systems, a new scheme was introduced, called a self-adaptive random early detection (SARED), which dynamically adjusts its drop pattern based on the current traffic load. This overcomes the limitation of traditional active queue management (AQM) [8]. A different study was conducted to investigate the age of information (AoI) and peak AoI (PAoI) metrics in multisource IoT systems [9], which analyzes three queueing disciplines for the status update process. Meanwhile, queueing network-based approaches were presented to analyze the consistency of consortium blockchain protocols and their performance [10].

The research papers mentioned above highlight the critical role of queueing systems with different models, motivating us to explore a more refined approach for adapting them to our system.

### 2.2. Reinforcement Learning for Sustainable Networking

In such a dynamic and complex environment, the powerful reinforcement learning AI method optimizes decision making, including wireless networks and queueing systems. Many existing research studies have proved the ability of RL to enhance traffic management and resource allocation and optimize network control policies, as in [11] for queueing networks, focusing on minimizing job delay and queue backlog. The simulations demonstrate their effectiveness in dynamic server allocation as well as the routing and switching scenarios. Similarly, RL has been used for service-rate control in tandem queueing systems, ensuring that the resources are managed efficiently while the end-to-end delay is guaranteed [12]. Advanced policy gradient methods, such as proximal policy optimization, have improved queuing network control [13], which supports the work in adaptive queue management for 6G systems.

Furthermore, a combination of graph neural networks and deep RL helps in optimizing network automation and resource management [14], which supports intelligent control for the upcoming generation of networks. Another work [15] proposes a deep RL-based active queue management scheme for managing traffic in IoT networks using fog/edge architecture. The results show that it outperforms other active queue management schemes in terms of KPIs, demonstrating its efficiency in congestion management. Generative diffusion models (GSMs) are considered a powerful tool for network optimization when combined with deep RL, where the researcher in [16] highlights their application in various network domains. Additionally, ref. [17] examined the use of learning techniques in optimizing queueing systems, discussing the connections between queueing theory and adversarial learning. Moreover, the researchers highlight the importance of service parameter estimation, decision estimation, and recent advances in RL. From the related research papers mentioned and others, we saw the effectiveness of RL and deep learning in optimizing network control and queueing management. We believe that by building on these advancements, our work further refines network optimization strategies. A closely related study [18] proposed a distributed deep reinforcement learning method for gradient quantization in federated learning within vehicular edge computing, targeting ultra-low latency and communication efficiency as key requirements for 6G systems. While their work focuses on distributed model training efficiency, our study extends the application of deep reinforcement learning to dynamic queueing decisions, addressing complementary aspects of adaptive network behavior.

### 2.3. 6G Promising Technologies

As the 6th generation of cellular networks emerges, integrating artificial intelligence becomes essential to optimize network performance and bring promising technologies. Therefore, many researchers are currently exploring ML techniques to improve the KPIs of the 6G. The integration of AI and ML in wireless networks was studied in [19,20] to lower costs, enhance performance, and enable automation and predictive analytics, which are essential for next-generation networks. Moreover, integrating deep RL can handle complex state and action spaces [21] and shows a better performance than conventional methods in simulating the UAV-IRS trajectory, trying to minimize mission time and energy consumption [22].

Nevertheless, researchers in [23] have examined the threshold of steady-state probabilities for stable queueing systems with long-range interactions (LRIs). Key techniques such as algorithm unrolling, graph neural networks, and federated learning are highlighted in [24] and applied to large-scale problems in 6G. While another paper [25] focused on how ML helps address challenges in resource allocation, task offloading, and mobility management for the upcoming 6G cellular networks. Another aspect was presented in [26], where they proposed an RL framework for optimizing channel access in IEEE 802.11 standards for mIoT using a probability dataset to improve resource allocation in the upcoming generation.

Digital twin technology faces challenges in 6G due to network complexity and scale [27]; however, generative AI offers solutions for modeling, synchronization, and slicing. Meanwhile, the evolution of mMTC and URLLC in 6G was explored by the researchers in [28], who discussed AI-based techniques, such as explainable AI and digital twin, to meet the required aspects of critical services. Nevertheless, other research works focus on integrating different AI methods in 6G and studying their contributions to enhance and overcome the challenges that 6G faces [29]. Different processing methods, such as the Empirical Mode Decomposition, were applied to the terahertz communication dataset to analyze it and extract some properties [30]. The integration of these useful advanced technologies into the upcoming cellular networks plays a crucial role in shaping them. These studies not only guide our understanding of the necessary technological advancements for 6G but also provide a solid basis for our research study.

### 2.4. Fundamentals of Queueing Theory and Its Applications in Wireless Networks

Queueing theory provides a rigorous yet intuitive framework for analyzing systems where multiple entities compete for limited resources. It helps in enabling the evaluation of key performance metrics, such as waiting times, blocking probabilities, system utilization, and throughput. It is essential to understand these metrics for designing and optimizing communication systems, particularly under dynamic conditions. The queueing system Kendall’s notation (A/B/C/K/m/Z) [31] is a widely used method for characterization, which succinctly describes the nature of arrivals, service time distributions, the number of servers, queue capacity, population size, and service discipline. Classic models, such as M/M/1, M/G/1, and retrial queues, have been extensively explored due to their analytical tractability and their ability to capture important real-world behaviors. The queueing theory helps wireless communications in assessing medium access control mechanisms, handling retransmissions, managing congestion, and optimizing resource allocation. Its relevance is heightened in emerging 6G networks, which pose new challenges, such as ultra-dense node deployments, highly directional terahertz communications, and the integration of AI-driven adaptive resource management. Queueing models serve as powerful tools to understand these complex dynamics and support the development of robust, efficient communication protocols.

Recent literature continues to underscore the significance of queueing theory in next-generation wireless systems. For instance, a comprehensive tutorial on mathematical modeling of 5G/6G millimeter wave and terahertz cellular systems emphasizes the role of queueing-theoretic approaches in capturing the dynamics of resource allocation and session-level performance in wireless networks [32]. Additionally, a detailed review of queueing models for communication systems highlights the necessity of advanced models to address bursty traffic and diverse quality-of-service requirements in modern networks [33]. Further studies have addressed specific challenges, such as network slicing and multi-tenancy. In [34], the authors propose a utility-driven multi-queue admission control strategy to handle heterogeneous service requests in network slicing scenarios. Meanwhile, ref. [35] investigates queueing-based service policies for wireless caching helper systems, explicitly modeling heterogeneous traffic with both cacheable and non-cacheable content types to improve throughput and user experience.

## 3. Methodology of the Retrial Queueing Systems M/M/1/K Model

We used the queueing system Kendall’s notation, a standardized way to describe queueing models. It provides a concise representation of their key characteristics. Kendall’s notation follows the format A/B/C/K/m/Z [31,36], where A is the arrival process (e.g., M for Markovian or Poisson arrivals), B denotes the service time distribution (e.g., M for exponential service times), and C is the number of servers. The parameter K indicates the system capacity, including both customers in service and the queue, and m represents the size of the calling population (often assumed infinite if omitted). Finally, Z is the service discipline, such as first-come-first-served (FCFS). By following this notation, the retrial M/M/1/K model describes a system with Poisson arrivals [31,36]. Exponential service times, a single server, and a finite buffer size make it well suited for analyzing many KPI factors in 6G networks, such as congestion and resource allocation.

Among all the queueing types that exist, we use a simple example of an M/M/1/K retrial queue as shown in Figure 1. M/M/1/K is considered a fundamental model that extends classical queueing systems by using the concept of orbiting customers. This concept allows the MTs who find the server busy to retry after a random delay instead of leaving the system or becoming a lost MT. The model used is relevant for such systems where blocked MTs do not immediately depart but make future attempts to be served. The ‘Free’ state in the queue means that there is at least one free location in the queue buffer. We chose this model to study 6G networks because it effectively captures the impact of blocked mobile terminal requests that reattempt service after a delay, which reflects real-world scenarios where, under dynamic network conditions, users experience temporary unavailability.

### 3.1. General Concept of the Model

In our simple example, the arrivals follow a Poisson process with rate λ, where the interarrival times are independent and exponentially distributed. Moreover, the service time follows an exponential distribution with a mean 1/μ, while we implement a single server and a finite queue size K. As we mentioned earlier, the blocked MTs enter an orbit and are retrieved at an exponential rate θ rather than leaving the system and being immediately lost.

While it is true that many other service time distributions, such as Gamma, Weibull, or log-normal, can model more complex or realistic service behaviors, the choice of exponential service times in our M/M/1/K retrial queue model is deliberate. The exponential distribution possesses a memoryless property, meaning the probability of service completion in the next instant is independent of how long the service has already been in progress. This property simplifies the mathematical analysis and allows for tractable, closed-form solutions of key performance indicators, which is crucial in providing fundamental insights into system behavior. Moreover, in many practical 6G scenarios involving random access and rapid service attempts (e.g., retransmissions, short control message handling), service times can be reasonably approximated as exponentially distributed due to their stochastic and memoryless nature. While more complex distributions could capture service time variability more accurately, they typically lead to models that require numerical or simulation-based analysis, which might obscure the fundamental system dynamics we aim to highlight. Hence, the exponential service assumption balances analytical tractability with sufficient fidelity for modeling and understanding the core mechanisms of 6G retry queues under study [37].

This retrial queueing model is useful in the upcoming 6G cellular networks, where many unique challenges are introduced in resource allocation and connectivity due to ultra-high frequencies and directional communication. The validity of assuming Poisson process arrival as a simplified example in such complex networks has been supported by multiple studies [38,39,40], where it was employed in IEEE 802.15.3c, IEEE 802.11ad, and IEEE 802.11ay/MiWEBA models. Other types, such as the Poisson Point Processes (PPP) [32,41] and Switched Poisson Process (SPP) [42], have proven their applicability in high network scenarios.

While alternative point processes, such as Cox, Hawkes, or determinantal point processes, provide advanced modeling of spatial–temporal dependencies, the Poisson process remains a scientifically sound and analytically tractable choice in this context. Its use is particularly justified due to its memoryless property and ability to capture the statistical independence of mobile terminal arrivals, key assumptions that align well with random access patterns and decentralized behavior in 6G THz networks. Furthermore, Poisson-based models support closed-form expressions for key performance metrics, such as blocking probability, access delay, and interference levels, which are essential for the theoretical design and optimization of next-generation MAC protocols. The superposition and thinning properties of Poisson processes also allow flexible modeling of aggregate user behavior and dynamic load conditions, which would be analytically intractable under more complex processes in early-stage evaluation [43].

Given this, our assumption that arrival in the queueing model follows a Poisson process is aligned with existing 6G cellular network models that operate on the THz frequencies, particularly in random access mechanisms, contention-based service requests, and user mobility scenarios. Since we can model the Mobile Terminals (MTs) arrivals, Access Point (AP) deployments, and interference sources as Poisson processes in such systems, the RQS is a natural fit for capturing the dynamics of blocked MTs retrials, intermittent connectivity, and access contention in 6G THz networks.

We acknowledge that the high dynamism of 6G THz networks, especially user mobility, bursty traffic patterns, and blockage effects, can cause deviations from a strict Poisson assumption. These factors may result in arrival correlations or short-term traffic bursts not fully captured by the memoryless nature of Poisson models. Nonetheless, the Poisson process offers a tractable and widely adopted abstraction that allows modeling average behavior and provides valuable insights into the system dynamics, particularly in access scenarios where aggregated user activity can approximate a Poisson distribution.

One of the scenarios for Poisson arrival is the adaptive directional antennas for terahertz frequencies (ADAPT) MAC mechanism [44], which is a MAC mechanism made for terahertz frequencies and compatible with the terahertz physical layer defined in IEEE 802.15.3d. The AP is placed in the center of tens of meters of cells, while the MTs are placed around the AP. The cell is divided into a fixed number of sectors to decrease the area of focus for the AP to increase the throughput and directional beamforming efficiency; therefore, the AP will be rotating periodically through these sectors, sending and receiving data. Since MTs attempt to access the AP dynamically, their arrivals can be effectively modeled as a Poisson process. This assumption aligns with the previously mentioned studies, the Poisson process assumption is relevant in scenarios where MTs independently transmit data whenever they enter an active sector, being random but statistically predictable patterns of arrivals at the AP.

However, when the MTs miss their opportunity to access the AP, they may need to retry access in a later sector rotation, introducing a retrial-based queueing dynamic. This end fits naturally into an M/M/1/K retrial queueing model, where the blocked MTs enter an orbit and attempt retransmission after a random delay. In ADAPT, missed transmission opportunities do not necessarily result in permanent loss, but rather it retries to transmit in the next AP’s rotation.

From what has been proven, we consider our RQS within a 6G network scenario, where high-frequency, directional, and intermittent access characteristics of THz communication networks require a retrial-based approach. Our queueing framework aligns with the characteristics of 6G scenarios involving stochastic MT distributions and dynamic access attempts. The beam misalignment, sectorized transmissions, and intermittent connectivity of 6G networks that can lead to delayed but eventual access opportunities are relevant with the assumption that blocked MTs enter an orbit and retry to access the AP at a later time. By leveraging the RQS model, we capture these fundamental dynamics, ensuring that the presented approach remains scientifically grounded and applicable to the upcoming cellular networks.

For our network scenario, the interarrival time between MTs has an exponential distribution with parameter λ. It shows that the arrival process is memoryless, and the probability of a new MT occurring in a small time interval dt is approximately as follows:(1)$$P_{arrival}\approx \lambda dt$$

Equivalently, the AP answering time has an exponential distribution with a mean 1/μ, where the probability that a service completion occurs in dt is(2)$$P_{service}\approx \mu dt$$

One has to note that these exponential distributions lead the model to be designed as a continuous-time Markov chain (CTMC), having transitions between states occur with rates dependent on the set parameters λ, μ, and θ.

### 3.2. Orbit and Retrial Mechanism

In the retrial queueing systems, MTs who find the queue full do not leave the system and are not considered lost MTs; in contrast, they enter a so-called orbit, as was mentioned previously, and retry to enter the system once again after an exponentially distributed delay with mean 1/θ. The probability of an MT retrying with a time interval dt is(3)$$P_{retry}\approx \theta dt$$

Therefore, we can say that the retrial process follows a memoryless property. The steady-state behavior of the system is analyzed by taking into consideration the number of MTs in the queue and the orbit, where the number of MTs in the queue at time t is noted as Q(t), the number of MTs in orbit at time t is noted as O(t), and S(t) is the number of served MTs at time t.

The model state transitions follow a so-called balance equation, which describes the probability of each state over time t as follows:(4)$$dP_{n}/dt=\lambda P_{n-1}-(\lambda +\mu )P_{n}+\mu P_{n+1}\;,\;\mathrm{for}\;1\leq n<K$$

In case the system is full (n=K), the balance equation incorporates retrial MTs as follows:(5)$$\mu P_{K}=\lambda P_{K-1}+\theta P_{orbit}$$

In Formulae (5), $P_{orbit}$ represents the probability of an MT being in the retrial state:(6)$$P_{orbit}=\lambda P_{k}/\theta$$

Formulae (6), guarantees that the system stays ergodic under the stability condition represented as the following:(7)ρ=λ/μ<1

This requires that the effective arrival rate of the MTs does not exceed the service rate, which will testify that the system will not grow indefinitely.

### 3.3. Performance Metrics

To evaluate the efficiency of the RQS model, we define some key performance metrics, starting with the average queue length ($L_{q}$) defined as follows:(8)$$L_{q}=\sum_{n=0}^{K}nP_{n}$$
where $P_{n}$ represents the probability of having n MTs in the queue. Next, the expected number of orbited MTs ($L_{0}$) is as follows:(9)$$L_{0}=\lambda P_{K}/\theta$$
which defines the average number of MTs in the orbit, meaning those who have a chance to retry for service. Nevertheless, we have the mean system time noted as W, which is the average time an MT spends in the system, taking into consideration the waiting time and the service time, and it is calculated as follows:(10)$$W=L/\lambda_{eff},\;\;\;\;L=L_{q}+L_{0}$$

L is the total number of MTs in the system. Finally, we have the AP utilization factor ρ that represents the fraction of time the AP is busy, as follows:(11)ρ=λ/μ

As we saw in the stability condition, Equation (7), the system stays stable as long as we keep ρ<1, meaning that the arrival rate of the MTs to the system is always less than the effective service capacity.

### 3.4. Implementation of the RQS Model and Simulation Results

We evaluate the performance of the retrial queueing model, as we first implemented it as the pseudocode as shown in Algorithm 1.

The model dynamics include the explained terms, MT arrival, queueing service, and retrial processes, which were implemented based on the described mathematical framework. In our simulation scenario, all the used parameters are presented in Table 1, where the key values, such as the MTs’ arrival rate (λ), the service rate (μ), the queue capacity (K), and the retrial rate (θ), are all specified. For the aim of better understanding the dynamics of the retrial queueing systems, and with the help of MATLAB R2024b, we plotted the evolution of served MTs (with red color), queued MTs (with green color), and orbited MTs (with blue color).
**Algorithm 1.** Retrial queueing system pseudocode.1. *// Input*2. λ: Arrival rate, μ: Service rate, θ: Retry rate, 3. time_sim: Total simulation time,4. k = Maximum queue length

5. *// Initialize the Simulation*
6. Initialize to zero:
7.  MTs_in_queue, MTs_served
8.  MTs_in_orbit, AP_busy
9. for t=1 to time_sim
10.  *// MTs arrival based on Poisson process*
11.  if rand() < (λ/time_sim)
12.   if AP_busy = False
13.    *// AP is free, serve the MT*
14.     AP_busy = True
15.    MTs_served = MTs_served + 1
16.   elseif MTs_in_queue < k
17.    *// AP is busy, add MT to the queue*
18.    MTs_in_queue = MTs_in_queue + 1
19.   else
20.    *// Queue is full, MT go to the orbit*
21.    MTs_in_orbit = MTs_in_orbit + 1

22.  *// Service completion*
23.  if AP_busy = True and rand() < (μ/time_sim)
24.   AP_busy = False
25.   if MTs_in_queue > 0
26.    *// Serve an MT from the queue*
27.    MTs_in_queue = MTs_in_queue − 1
28.    AP_busy = True
29.    MTs_served = MTs_served + 1

30.  *// MTs retry from orbit*
31.  if rand() < (θ/time_sim) & MTs_in_orbit > 0
32.    if AP_busy = False
33.     AP_busy = True
34.     MTs_served = MTs_served + 1
35.     MTs_in_orbit = MTs_in_orbit − 1
36.    elseif MTs_in_queue < k
37.     // Move MT from orbit to queue
38.     MTs_in_queue = MTs_inqueue + 1
39.     MTs_in_orbit = MTs_in_orbit − 1
40.    else
41.     *// MTs remain in orbit (queue is full)*
42. end for

The visualization illustrates how the MTs move through the system from the arrival point to the service point, and how the queue and orbit states evolve.

As shown in Figure 2, the served MTs reveal a gradual increase over the simulation time, demonstrating the ability of the AP to serve at a steady but limited rate. In the beginning, the system serves a few requests, but as time progresses, the service rate adjusts to serve more requests. Nevertheless, the serving was not instantaneous, but followed a stepwise pattern, denoting that the used model operates under constraints that limit MT processing.

Concerning the queued MTs, we can say that they fluctuate over time, increasing with the increasing number of MT requests and decreasing as they are served. Some periods of stability are observed, meaning that the queue length remains unchanged, indicating that the arrival rate of the MTs and the service rate are momentarily balanced. However, there are instances where the queue state grows due to an influx of new arrivals, as shown in the figure, which can lead to delays and congestion if the service rate does not compensate for the augmented load.

The orbited MTs in the figure remain at a low level at the beginning of the simulation; as soon as time progresses, they increase when the queue reaches its capacity. This denotes that once the system becomes saturated, the newly arrived MTs are forced into the orbit, waiting for their chance to join the queue. Over time, they grow, reflecting the challenges of integrating orbited MTs into the queue in an efficient manner.

By the end of the simulation time, it had successfully served 28 MTs in total, 5 MTs remained in the queue waiting to be served, and 10 MTs were still in the orbit, unable to rejoin the queue. The results show the challenges and limitations of the simulated RQS model in a 6G framework context. The fact that there are remaining MTs waiting to be served in both the queue and orbit shows that the RQS simulated model under the given configuration cannot fully serve all received requests within the simulated time frame. The findings align with the real-world 6G scenarios, where high-frequency communication is susceptible to sectorized transmissions, blockages, and intermittent access, resulting in retransmission delays and service inefficiencies.

To optimize the performance of the system model used, we propose to integrate reinforcement learning to improve decision making, reduce congestion, improve the service rate, and minimize overall waiting times. By doing so, the system can proactively adjust to network conditions, traffic variations, and user mobility, making it more robust for the upcoming 6G and enhancing its KPIs. A detailed discussion of the implementation will be explained in the following section.

## 4. Methodology of Reinforcement Learning for Efficient Queue Management

One of the most famous and powerful AI frameworks for decision making in dynamic systems is reinforcement learning, thanks to the role of its key element, the agent, which interacts with the environment to learn and maximize a cumulative reward. Integrating RL has been widely adopted in dynamic wireless networks for optimizing many factors, such as traffic scheduling, adaptive access control, and resource allocation, making it a promising technology for the next generation of 6G networks. We believe that by integrating such a powerful tool into the 6G network-based RQS model, we can improve its key performance indicators, such as system efficiency, overall balance, and fairness. Moreover, it can help in enhancing its adaptability to dynamic network conditions and reduce service delays.

### 4.1. RL Q-Learning and Deep Q-Network

Among the most widely used RL techniques are Q-learning and Deep Q-Networks (DQNs). In the next subsections, detailed descriptions of these methods are presented.

#### 4.1.1. RL Q-Learning and Q-Table

Q-learning is a value-based RL method that enhances the agent’s performance and enables it to learn an optimal policy for managing the overall system by influencing the AP [45]. The Q-learning method relies on one of the most effective decision-making equations, the *Bellman equation* [46]. It formulates the relation between the value of a state-action pair and the expected future rewards, which makes it a unique way. The Q-value update rule is as follows:(12)$$Q(s,a)\leftarrow Q(s,a)+\alpha [r+\gamma \underset{a^{\prime }}{\max}Q(s^{\prime },a^{\prime })-Q(s^{\prime }a)]$$

In Equation (12), Q(s,a) represents the current value of taking action a in state s, and α is the learning rate, meaning the controls how much new information overrides the past knowledge. r and γ represent the rewards received for the action and the discount factor, respectively. The discount factor determines the importance of future rewards. Finally, $\underset{a^{\prime }}{\max}Q(s^{\prime },a^{\prime })$ is the highest estimated value of the next state s′. This repetitive process allows the agent to stabilize its decision making over time, improving the efficiency of the system by maximizing the total rewards.

The learned state-action pair of Q-learning is stored in a so-called Q-table [47]. This end is a matrix where each row represents a state s In the system, each column represents an action a that the agent can take, formulating a cell Q(s,a) that represents the expected reward for taking action a in state s based on past learning. During learning, the Q-table is updated based on the *Bellman equation*. Over multiple iterations, the Q-values gradually stabilize as the agent refines its knowledge, and the table converges to optimal values, enabling the agent to make the best decisions based on each state.

Despite the Q-learning benefits, it suffers from inherent limitations, especially for complex problems. The curse of dimensionality is one of its primary drawbacks. Q-learning relies on tabular Q-value representations, where the number of state-action pairs grows exponentially with the size of the state and action spaces [48]. This aspect makes Q-learning impractical for large or continuous state spaces. Another point is that for high-dimensional situations, the agent needs an extensive amount of interaction with the environment to refine and converge the Q-tables’ values, leading to slow learning, high computations, and inefficient exploration. This is particularly important in the case of RQS’s KVIs. There are other limitations that Q-learning suffers from, such as slow convergence in an environment with sparse rewards. In non-stationary environments, the agent assumes that the environment’s dynamics stay unchanged during training; meanwhile, in many real scenarios, the reward and the transition probability evolve over time.

To overcome these challenges, advanced RL techniques have been developed, which allow for environment adaptability, better scalability, and generalization, making it suitable for dynamic and complex problems. We discuss this in the following sub-section.

#### 4.1.2. Deep RL for Queue Optimization and Critic-Based Methods

The Q-learning method suffers from limitations as mentioned, especially for dynamic systems. The deep Q-network method overcomes these limitations by integrating deep neural networks as function approximators. Instead of following the Q-table that grows exponentially with the state–action space, DQN utilizes a neural network to approximate the Q-values for each state [49]. The method used helps the agent to handle complex systems with large and continuous state spaces effectively. Deep Q-network uses experience replay, where past experiences, such as the state, action, reward, and next state, are stored in a buffer and randomly sampled during training, which improves the stability of learning. In addition to what has been mentioned, DQN employs a target network that updates copies of the main Q-network periodically and stabilizes learning.

One of the methods that belongs to the border class of policy-based reinforcement learning is the critic-based methods, which learns the explicit policy rather than a value function [50]. The agent in these methods consists of two components: the actor and the critic. The actor determines the policy by mapping states to actions, and the critic evaluates the actions taken by estimating the value function. Critic-based methods have several benefits; they are more stable in continuous action spaces, and they can adapt better to non-stationary environments, as they do not require discretization.

### 4.2. Integrating RL Framework into the RQS Model

In our design, we chose RL to act as an upper-layer controller that dynamically enhances the performance of the AP based on real-time observation from the environment. The integration of RL in the RQS is shown in Figure 3 and in Algorithm 2, which has the following structure: the environment consists of the input source (λ), the queue (K) and the orbit (∞). The agent observes the system state, including queue size and AP availability. Based on the observation, the agent selects an action:

Action=0: Do nothing, let the system operate normally.Action=1: Force the AP to serve one MT.

The AP must execute the action, impacting the environment state, where the reward is assigned based on the performance, encouraging actions that improve one of the KVIs, system efficiency.

We define the reward function as shown in Algorithm 2 and Equation (13). The reward equation is designed to balance the trade-off between serving more mobile terminals and preventing excessive queue build-up. α is considered a scaling factor to encourage serving more MTs during the simulation time, reinforcing actions that improve the throughput and decrease congestion. Δ is the change in the number of MTs after taking an action. A higher value of Δ(served MTs) increases the reward, influencing the agent to prioritize service efficiency.

On the other side of Equation (13), β represents a weight that regulates the penalty for increasing queue length. A higher queue length increases the penalty and decreases the reward, discouraging actions that might lead to network congestion and inefficiencies.

By adjusting scaling factor α and weight β, we control the agent’s decision making, ensuring the balance between maximizing throughput and preventing service bottlenecks. The used mechanism allows the reinforcement learning agent to dynamically adapt its policy based on real-time network conditions and optimize the 6G-RQS model for reaching higher efficiency and better resource utilization.(13)Reward=α·Δ(served MTs)−β·Δ(queued MTs)

For implementing the RL in our 6G-RQS model, we utilized DQN, which extends traditional Q-learning, as was explained previously. The state space used in our DQN-RL model is based on two parameters: the queue length and the AP status (busy/free). This choice was made to maintain a simple and interpretable model structure, particularly suited for evaluating the core dynamics of the retrial queueing system. We tend to keep it simple because adding many parameters to the state space in reinforcement learning can lead to the curse of dimensionality, where the state-action space becomes too large to explore efficiently. This increases training complexity, slows convergence, and can result in overfitting or unstable policy behavior, especially with limited training episodes or dynamic environments. We consider this representation as a foundational step toward more advanced models.

The layered design of the critic neural network is shown in Figure 4 to effectively capture the complex relation between the state and the optimal actions. The input to the network is the state representation of the 6G-RQS system; next, we have the first fully connected layer that consists of 24 neurons, capturing the high-dimensional features of the state space.
**Algorithm 2.** Reinforcement learning integration pseudocode.1.*// Define Environment**//   contains the RQS simulation-based inputs, state, and reward*2.*// 1. Simulation-based Input*3.λ: Arrival rate, μ: Service rate, θ: Retry rate, 4.time_sim: Total simulation time, 5.K = Maximum queue length

6.*// 2. State (2D vector)*7.Number of MTs in the queue8.AP status (busy/free)

9.*// 3. Reward*10.R=α·(Δserved MTs)−β·(ΔMTsinqueue) 

11.// Define observation12.Number of MTs in the queue13.AP status (busy/free)

14.*// Define Action*15.if Action = 016.  Agent does nothing17.else18.  Agent forces AP to serve one MT


// Update State19.if MT arrives:20.  if AP free 21.   AP serves MT22.  elseif queue < K 23.   MT joins queue24.  else25.   MT lost26.if AP finishes serving27.  AP becomes free28.  If queue not empty29.   Serve next MT from queue30.if retry occurs and lost MTs > 031.  if queue < K32.   Move one lost MT to queue

We used the rectified linear unit (ReLU) activation function as the third layer in the NN to introduce non-linearity, ensuring the network can model complex decision boundaries.

The fifth layer is again a fully connected layer, this layer further refines the learned features and contributes to the depth of the network. It follows that the second ReLU activation function enhances non-linearity and stability. Finally, a third fully connected layer is an output, which provides the Q-values for each possible action that the agent will use to select the optimal action for a certain state based on the highest predicted reward.

Table 2 presents the key hyperparameters used for the DQN-based critic network. These parameters are selected to ensure stable and efficient learning, balancing convergence speed and generalization performance. These hyperparameters play a crucial role in refining the performance of the DQN model, ensuring that the critic network effectively learns optimal actions to enhance 6G-RQS system efficiency.

### 4.3. Implementation of the RL-6G-RQS Model, Simulation and Performance Evaluation

In this subsection, we present 6G-RQS simulated scenarios based on different parameters and their performance. Table 3 presents the parameters used in the simulated model, where we systematically varied three key factors to analyze the system’s performance under different conditions. The first parameter is the arrival rate λ, which we adjusted to six different values to examine its impact on the system behavior and congestion level. Additionally, we modified the queue size K across four different cases to assess how varying buffering capacities affect service efficiency. Moreover, scaling factor α in the reward function also varied across five distinct values, always between 0 and 1, to investigate its influence on the RL agent’s decision-making process. This results in a total of 120 simulated scenario cases (6 × 4 × 5). The simulation duration of 100 time units was selected to provide a balance between computational efficiency and capturing sufficient queueing dynamics for analysis. This timeframe allows the reinforcement learning model to effectively converge and reflect the system behavior under varying traffic conditions. Extending the simulation length significantly would increase computational complexity and training time, potentially hindering rapid evaluation and iteration during the development phase. The chosen duration aligns with common practice in wireless network simulation studies focused on latency-sensitive applications, where timely decision making and responsiveness are critical. In real-life scenarios, traffic patterns can vary widely depending on the application and environment, with edge devices often requiring rapid adaptation within short timescales. Thus, our simulation setup aims to realistically model these short-term dynamics typical in mission-critical wireless networks.

The queue size K varies across values 10, 15, 20, and 40 to emulate constrained edge environments where latency-sensitive services, such as URLLC and mission-critical IoT, operate under tight buffer limitations. This range reflects the practical trade-off between maintaining low latency and ensuring manageable computational complexity during training. In real-world wireless systems, buffer capacities vary widely depending on the deployment scenario. While backbone and core components may support buffer sizes in the thousands, edge nodes, such as small cells, base stations with mobile edge computing capabilities, or vehicular network units, often operate with significantly smaller buffer spaces to minimize queuing delay and jitter. For example, 3GPP specifications for URLLC traffic profiles emphasize minimal buffering to meet sub-millisecond latency targets. By exploring queue sizes within this low range, the model captures the behavior of these delay-sensitive systems while still allowing the analysis of buffer size impact on system performance.

To assess the impact of reinforcement learning on system dynamics, we analyzed different performance metrics over the learning process and across 10 training episodes. Figure 5 and Figure 6 illustrate system behavior under two different fixed queue sizes (K) and reward parameters (α,β), while considering all six different arrival rate λ scenarios. The weight β=1−α ensures that the reward function always maintains a balanced trade-off between different objectives, such as the KPIs throughput and delays, without arbitrarily amplifying or diminishing their combined influence. By doing so, the total weight sum is constant (α+β=1), preserving the relative importance of each term while allowing controlled adjustments through α. Each figure has three subplots that show the evolution of three metrics: the number of served MTs, the number of queued MTs, and the number of orbited MTs throughout the simulation time.

By analyzing these trends, we could evaluate the model’s ability to optimize the AP’s service efficiency, reduce congestion, and balance the resource allocation in a 6G-driven queueing model.

Figure 5a represents the number of served MTs for a queue size of 20, scaling factor of 0.1, and reward penalty weight of 0.9 under six different arrival rate scenarios. The plot spans 10 episodes, and each episode consists of 100 time unit steps, exporting the system’s performance across the overall learning time. One can notice the gradually increasing trend in the number of served MTs from episode to episode for all arrival rate cases, reflecting the improvement of the system’s performance as the RL agent learns and adapts to the environment.

The improvement proves the agent’s ability to optimize decisions over time, gradually improving its understanding of the queueing dynamics. It is also important to highlight that while the training process of the RL agent may require substantial computation, the inference phase (used during actual decision making in real-time) is significantly faster. The inference step consists of a forward pass through the trained DQN, which takes less than a millisecond on typical edge AI hardware, such as those equipped with lightweight GPUs or AI accelerators. This ensures that the DQN-based decision making meets the ultra-low latency requirements of 6G URLLC, making the proposed solution viable for real-time applications in THz-band 6G environments.

However, in certain cases, there are periodic decreases in performance, where the number of served MTs drops for some time during specific episodes. This behavior can reflect the inherent nature of the RL, where the agent tries to explore other strategies in the early stages of learning, leading to suboptimal actions before convergence to better policies. However, this behavior is temporary because the number of served MTs consistently increases in subsequent episodes, demonstrating the effectiveness of the RL algorithm in balancing exploration and exploitation, leading to an improvement in system performance across simulation periods. We can say that this behavior is indicative of the agent’s continuous learning process, progressively adjusting to the system dynamics.

Figure 5b illustrates the evolution of the queued MTs across 100 time steps repeated for 10 episodes under the same scenario. Initially, during the first steps, we can observe that the queue increases until it becomes full, particularly for higher arrival rates (λ=70,90,100). This behavior is expected, as the agent has not yet adapted to the environment and has not yet learned the optimal policy for managing the requests and resource allocation. As the agent interacts more with the environment and refines its decision making over episodes, we can notice a decrease in the number of queued MTs.

For certain arrival rates, the queue eventually reaches zero, indicating that all MTs were served, which demonstrates the agent’s ability to effectively balance the system’s load and prevent excessive buildup of queueing MTs. In contrast, one can observe that when the queue becomes full at any point, a corresponding rise in the number of orbited MTs is seen. However, this was expected to happen, because the MTs who cannot join the queue are sent directly to the orbit rather than allowing to become lost (see Figure 5c).

The trends observed for served MTs, queued MTs, and orbited MTs for the previous scenario (α=0.1) are also evident for α=0.9 (see Figure 6). However, one noticeable difference exists between the two scenarios, which is the faster increase in the number of served MTs for α=0.9. This can be attributed to the higher scaling factor of the reward function, which influences the agent to maximize the number of served MTs rather than focusing on managing the queue size effectively. This higher scaling factor leads the agent to take actions that directly reduce the number of queued MTs, but might not take into account the long-term consequences of this approach. While the agent focuses more on improving the service rate of the AP, it might neglect queue management. This approach highlights the trade-off between the scaling factor in the reward function and the agent’s ability to optimize both the served MTs and the queue dynamics.

Moreover, we visualize the total number of served MTs per episode for two different scaling factor values, α=0.3 and α=0.9 (see Figure 7). The plot shows that in cases where the scaling factor α=0.3, there is a noticeable drop in the number of served MTs, particularly for higher arrival rates. The drops are characterized by high fluctuations in performance. This behavior suggests that the agent’s decisions may lead to suboptimal actions, mostly during the early learning stage. This could be due to the low reward scaling factor, which causes the agent to be less motivated to focus on enhancing the serving performance and more inclined to balance the system in ways that take time to achieve optimal performance. On the other hand, for α=0.9, the total number of served MTs shows a stable and gradual progression compared to the other scaling factor scenario, with a slight decrease in performance observed occasionally.

These smaller drops may indicate that if we set a high scaling factor for the reward, we encourage the agent to optimize the number of served MTs over time while maintaining better control over fluctuations. In general, the agent can adapt effectively under various circumstances and avoids large performance drops that are observed in the lower α scenarios. We can conclude that the choice of the scaling factor plays a crucial role in the agent’s ability to stabilize the system’s performance and manage trade-offs between serving MTs and system dynamics.

To evaluate the performance of the model and analyze the relationship between the components of served MTs, we performed singular value decomposition (SVD) on the overall learning time series of the served MTs for all cases. The SVD of matrix A is given by(14)$$A=U\;S\;V^{T}$$
where U is an m×m an orthogonal matrix containing the left singular vectors, S is an m×n diagonal matrix with singular values, and $V^{T}$ is an n×n orthogonal matrix containing the right singular vectors. For our model, we have m=10 episodes and n=100 simulation time for each simulation case. For dimensionality reduction and retaining the most significant components of the matrix, we selected the rank r=2, meaning that we chose the top two singular values to approximate the original matrix because these two singular values give over 90% of the information content of all singular values. The reconstructed matrix is calculated as follows:(15)$$A_{r}=U_{r}\;S_{r}\;V_{r}^{T}$$
where $S_{r}$ contains only the top r singular values, and $U_{r}$ and $V_{r}$ are the corresponding reduced matrices. To quantify the approximated error, we calculate the root mean square error (RMSE) between the original matrix of served MTs and the reconstructed $A_{r}$.

Additionally, for further analysis, we extracted the diagonal of the matrix S, and we plotted its second value versus its first value, which represents a comparison between the second and first singular values. This analysis gives us the right to explore how the most dominant singular values correlate with each other and whether the reduction to two components could effectively capture the features of simulated scenarios. By making this analysis, we not only study the quality of the served MTs’ matrix but also gain insights into the data structure and check the key factors that influence the performance of the system under varying configurations.

The first and second singular values extracted from the matrix S is represented as a scatter plot for all 120 simulated cases (see Figure 8). Interestingly, we could observe that different arrival rate scenarios naturally formed distinct clusters, resulting in six well-defined groups corresponding to the six arrival rate values used in our simulations.

One must note that the fifth and sixth clusters, which correspond to arrival rates λ=90 and λ=100, appear close to each other in the singular value space. This could suggest that the system exhibits similar structural characteristics under these high-load conditions, likely due to the saturation effects of the queueing system as the arrival rate approaches its upper limit. The found clustering behavior proves the strong impact of the arrival rate on the learned policy and system dynamics. Nevertheless, the distinct clusters reinforce the effectiveness of SVD in capturing the dominant patterns within the served MT matrix, offering valuable insights into the fundamental structure of the learned 6G-driven RQS system dynamics.

We also plotted the calculated RMSE for all 120 simulated cases to assess the reconstruction accuracy of the SVD-based approximation. Again, we observe a clustering effect in the RMSE values (see Figure 9), which appears to correlate with the arrival rates used in the simulations. The higher arrival rates tend to exhibit larger RMSE values, likely due to increased system congestion and variability in decision making as the agent learns to manage the queue and orbit more efficiently. Moreover, this indicates that for a higher arrival rate, the first two singular values are not enough, and we need more to approximate the original matrix. In contrast, lower arrival rates result in lower RMSE values, indicating more stable system behavior with fewer fluctuations in the number of served MTs.

We performed the SVD analysis again, this time on the combined matrix of orbited MTs and queued MTs, following the same methodology applied previously for the served MTs. We then calculated the RMSE between the original and reconstructed matrices using reduced-rank approximations r=2. Unlike the served MT case, the result for both the scatter plot of singular values (see Figure 10) and the RMSE distribution (see Figure 11) did not exhibit a clear clustering pattern.

This indicates the irregular behavior of the queued and orbited MTs and suggests that they are less structured, making it harder to differentiate scenarios based on SVD in this case. A possible explanation for this behavior difference is that the served MTs directly reflect the agent’s learning and decision-making process. In contrast, the queued and orbited MTs are more dependent on system congestion and queue dynamics, which fluctuate more unpredictably across different scenarios. The finding highlights the usability of SVD in capturing patterns of served MTs; however, it might be less suitable for decomposing the queued and orbited MTs due to their chaotic nature.

The overall results demonstrate the usability of integrating DQN-based RL in the next 6G and especially in RQS scenarios, significantly optimizing system efficiency and adaptability. The increased number of served MTs across episodes shows that the agent can learn and optimize decision making over a certain time. Moreover, the reduction of the queued and orbited MTs reflects the balanced resource allocation even under high arrival rates. The ability of the model to adjust the dynamic nature of the environment based on real-time observations underlines the potential of RL approaches in future wireless networks and proves its powerful uses for 6G cellular communication systems, promising improved fairness and greater scalability.

### 4.4. DQN Performance Evaluation in 6G-RQS Environment

To evaluate the performance of the DQN in optimizing the 6G-DQN, we analyzed the evolution of the Q-value over time. To be more specific, at each time step t within an episode e, we extracted the Q-values corresponding to all possible actions. Based on the DQN decision rule, action $a_{t}$ was selected to conform to the maximum Q-value:(16)$$a_{t}=arg\;\underset{a\in A}{\max}Q(s_{t},a,\theta )$$

$Q(s_{t},a,\theta )$ is the estimated Q-value for state $s_{t}$ and action a given the neural network parameters θ. The set of possible actions is represented by A, and $a_{t}$ is the action selected at time step t.

Given that we have 120 different simulated cases, each generating a series of 1000 length (i.e., 100 simulation time steps per episode over 10 episodes), it might be impractical to directly visualize them or try to compare between. To address this, we analyze the most frequently selected action within each episode. Specifically, for each time step e, we calculate the mode of the chosen actions over the 100 time steps. Therefore, we could interpret DQN’s learning behavior and optimization performance in the 6G-RQS model.

We visualize the most frequently decided action per episode by using a stacked plot representation (see Figure 12). We construct a stacked plot where the *x*-axis represents 24 cases, corresponding to the combination of six arrival rates λ and four queue sizes K (i.e., 6 × 4), and the *y*-axis represents 10 episodes, progressing from top to bottom, illustrating the evolution of the agent’s decision over time.

This results in five stacked plots, each corresponding to a different scaling factor α of the rewards function.

By the beginning of the learning process (upper sections of the plot), we could observe a structured pattern where the decision-making process exhibits four distinct periods, representing four queue sizes used in the simulation, each period has six cases representing the six arrival rates λ.

This finding indicates that in the initial learning stage, the agent’s policy is still developing, and its behavior is influenced by the predefined queue configuration rather than adaptive learning. As we progress further down the plot, meaning later episodes, thus a more involved learning stage, we observe that these distinct periods begin to be refined based on the states and cases.

This suggests that the DQN-based agent has gradually adapted to the dynamic nature of the 6G-RQS environment, learning to make more context-aware and optimized decisions rather than being rigidly influenced by static queue configurations. This adjustment is consistent across both presented cases and the other three scaling factor cases as well, reinforcing that the learning process is effective for different values of α.

We further analyzed the decision-making trends of the DQN-based 6G-DQN model, as we visualized the most decided actions throughout all learning episodes using a heatmap representation. We can see in all heatmaps the following: the columns correspond to the four different queue sizes K, the rows represent the six arrival rates λ, resulting in a 24-box grid (i.e., 4×6), where each box indicates the most chosen action for that specific scenario. Since the most used action varies depending on the reward scaling factor α, we generate five separate heatmaps, presented in Figure 13. The results demonstrate that for arrival rates between 20 and 90, the most frequently selected action is action 1 (‘do nothing’) across almost all cases, with only a few exceptions. Notably, we saw that during these simulated cases, the performance of the system was enhanced, characterized by an increasing number of served MTs with a decreasing number of queued MTs, along with efficient resource utilization, ensuring that the AP operates optimally without any necessity to force the servicing. This finding suggests that the DQN agent has successfully learned to regulate the queue dynamics and only forces the AP to serve when it is necessary to maintain an optimal system rate. The model effectively identifies the right moments to allow natural service progression versus when to take action, thereby preventing unnecessary congestion and ensuring high throughput.

In contrast, for the highest arrival rate case (λ=100), the agent consistently favors action 2 (‘force to serve one MT’) as the most frequently chosen action across all queue sizes. This result can be attributed to the fact that at extremely high traffic loads, the natural service rate becomes insufficient to prevent excessive queueing and orbiting. To mitigate queue overflow and maintain stability, the agent chose action 2, forcing service operations, ensuring that MTs are processed at a faster rate to prevent degradation of system performance.

For the critic NN, we mentioned that we have used a fully connected output layer containing 24 neurons. After the simulation is finished, the output layer produces compressed Q-weights for the overall Q-values, resulting in 24×2 (24 neurons as output and two actions) for each simulation case. Thus, we again determined the most used actions, but this time based on the agent’s experience after the simulation is finished.

We compared the Q-weights for each neuron output and selected the action with the maximum weight, then extracted the most used action across all simulated cases. The results were then visualized across five different reward scaling factors (α) using histogram representations (see Figure 14). This approach allows us to extract the learned decision-making patterns after the training phase is complete, providing insights into the agent’s policy adaptation across different scenarios.

One important aspect to highlight in Figure 14 is that for each α case, the sum of the red bars (action 1: do nothing) and the blue bars (action 2: force to serve) always equals 24 cases. For the majority of cases across all scaling factors, action 1 was selected most frequently. This reflects that the DQN agent has effectively learned to maintain system stability by selectively choosing when to intervene, rather than constantly forcing service that leads to unnecessary energy consumption. Thus, we can say that the agent has adapted to the system dynamics, identifying optimal moments to allow natural service progression while ensuring energy efficiency. Interestingly, in approximately four cases per α instance, we observe a higher selection of action 2 (‘force to serve’). Based on earlier analyses, these cases predominantly correspond to high arrival rate values λ=100. This behavior aligns with the agent’s learned policy, where at a moderate to low arrival rate, the system operates effectively without frequent intervention, allowing for smooth queueing and processing. At higher arrival rates, natural service progression is insufficient to prevent queue overflow and increased MT orbiting; in such cases, the agent actively forces service operations to maintain performance and prevent system degradation.

### 4.5. Discussion of the RL DQN Simulation Findings

The research study findings demonstrate that integrating DQN reinforcement learning into the 6G-RQS model significantly enhances system performance by dynamically adapting service decisions based on network conditions. We could see that the agents’ learned policy effectively balances serving MTs, queue management, and resource allocation, ensuring optimal performance while minimizing unnecessary energy consumption. The key observation from the research is that the agent efficiently adjusts to different traffic loads, learning when to prioritize service (high λ), and when to conserve resources (low λ). On the other hand, we have the singular value decomposition analysis, which further highlights distinct patterns in system behavior, particularly in served MTs, where traffic rates naturally form clusters, an insight that could aid in traffic prediction and resource optimization.

Furthermore, to highlight the effectiveness of the RL-enhanced model, we conducted a comparative analysis between the basic RQS (without RL) in Section 3 and the DQN-RL-integrated RQS in Section 4. The basic RQS, under the same simulation settings, served only 28 MTs, with five remaining in the queue and 10 in the orbit. This illustrates congestion and inefficiency in completing service requests within the simulation timeframe. In contrast, we could see that the DQN-RL-based system significantly improved service rate, reduced queue size, and minimized orbiting MTs, showing that the RL agent effectively adapts to network dynamics and improves system throughput.

These contributions align with the upcoming 6G cellular network goals of autonomous, intelligent, and ultra-efficient communication systems. The integration of RL in such complex and dynamic applications would overcome many challenges that the 6G faces, enhance URLLC, improve energy efficiency, and enable adaptive service mechanisms. Nevertheless, this integration is valuable for operating in the THz spectrum, where directional communication, high path loss, and dynamic beamforming challenges require intelligent MAC-layer adaptation.

Further investigation might analyze the effect of the retrial rate of the 6G-RQS model, as it plays a crucial role in determining system congestion levels and the efficiency of reattempting service for orbiting MTs. We could also explore multi-agent reinforcement learning (MARL) to enable decentralized decision making for dense ultra-dense network deployments, such as 6G networks. The MARL method could be useful for such scenarios that use multiple APs and MTs and interact dynamically; by doing so, the system could adapt itself under various conditions, mobility patterns, and interference conditions, ensuring seamless connectivity and efficient spectrum utilization in 6G-THz environments. Future validation with real-world datasets and hardware testbeds will be essential to confirm the scalability and robustness of the proposed model.

## 5. Conclusions

The study presented explored the integration of RL into the 6G-RQS model to optimize queue management in dynamic wireless environment scenarios. The regular RQS model suffers from many challenges, especially under high demands. We proposed to use the DQN approach to enhance system performance by increasing the number of served MTs, reducing queue congestion, and managing retrial processes. The findings revealed that the integration of such a powerful tool adapts to varying traffic conditions, learns optimal service policies, and balances energy efficiency with fair resource allocation. Singular value decomposition identifies structured patterns in served MTs, underlining the system’s learning process and decision-making trends. Moreover, the histogram and heatmap analyses of agent decisions confirmed that the RL effectively determined the best moments to force service while ensuring overall system efficiency. The results suggest that RL-driven queue management could absolutely improve future 6G networks and help in handling high-density traffic while improving fairness in resource distribution. Nevertheless, the framework is highly relevant for THz communications, where we find intelligent scheduling is a must due to its unique propagation challenges.

Future work will investigate the impact of the retrial rates, integration of multi-agent learning approaches, and the use of THz simulated data for validation to extend the proposed model’s applicability.

## Figures and Tables

**Figure 1 sensors-25-03621-f001:**
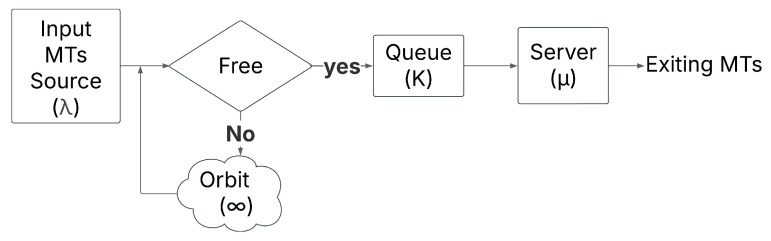
Retrial queueing system model.

**Figure 2 sensors-25-03621-f002:**
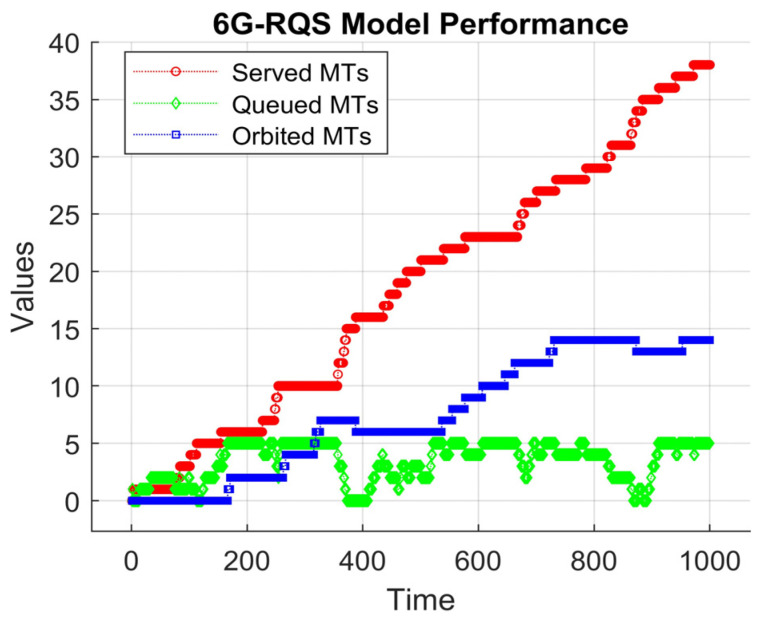
6G retrial queueing system model performance.

**Figure 3 sensors-25-03621-f003:**
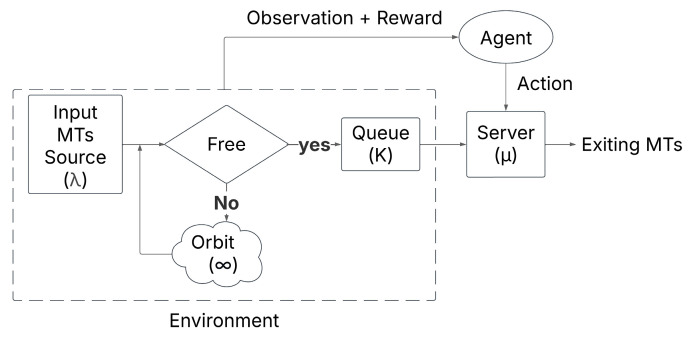
Reinforcement learning integration in the retrial queueing system model.

**Figure 4 sensors-25-03621-f004:**
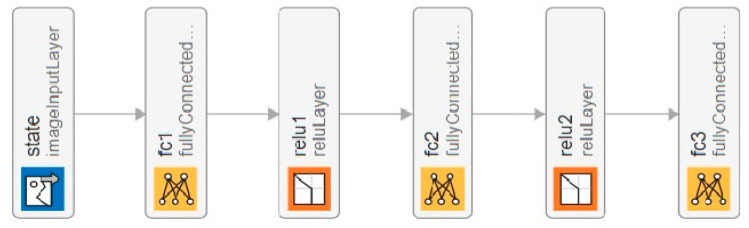
Design of the critic neural network.

**Figure 5 sensors-25-03621-f005:**
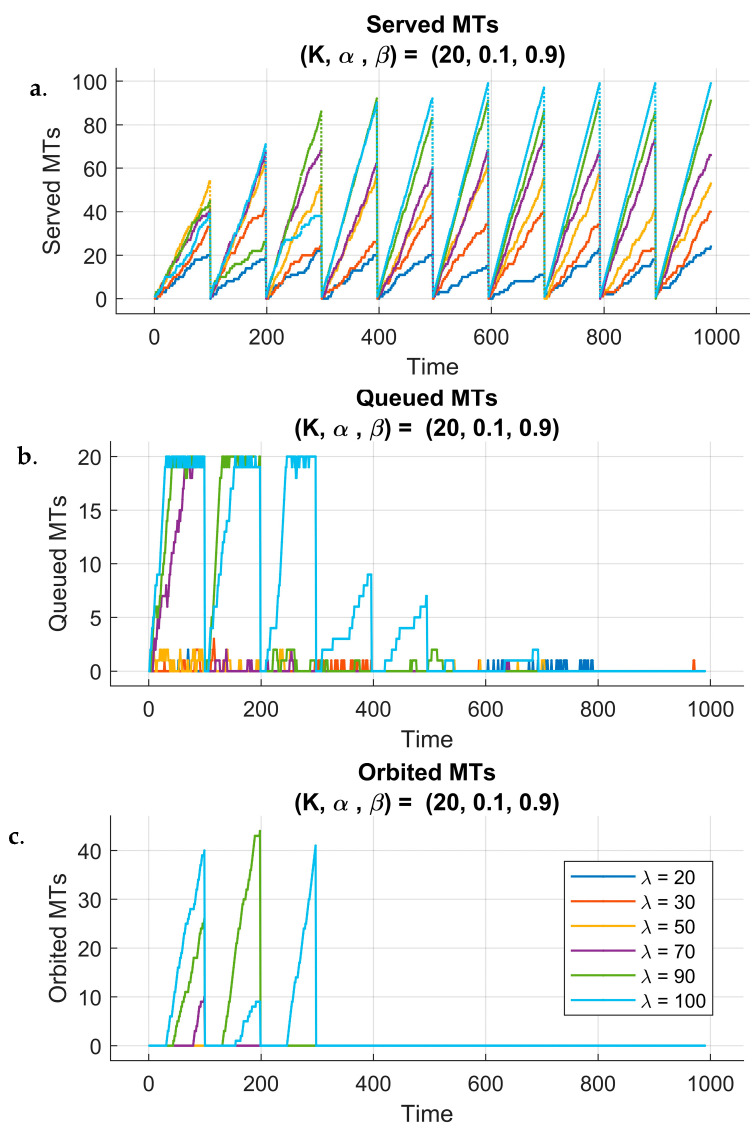
Performance of 6G-RQS with RL integration: served (**a**), queued (**b**), and orbited (**c**) MTs for (K,α,β) = (20,0.1,0.9).

**Figure 6 sensors-25-03621-f006:**
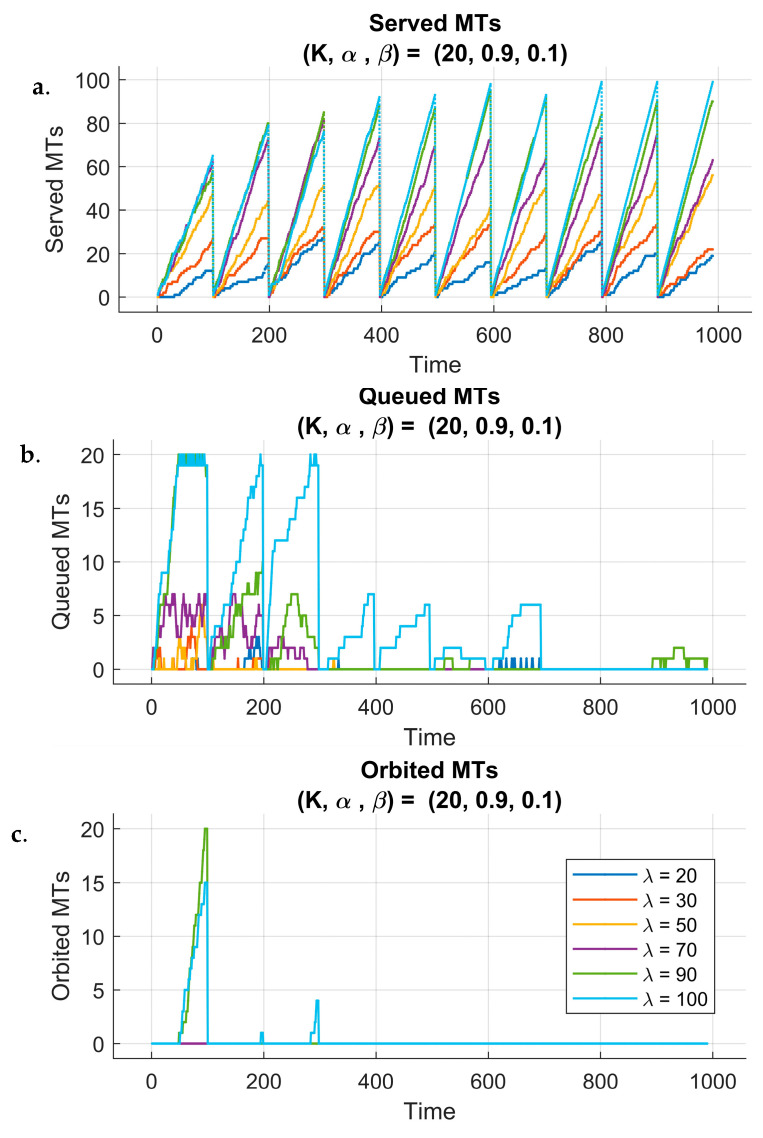
Performance of 6G-RQS with RL integration: served (**a**), queued (**b**), and orbited (**c**) MTs for (K, α, β) = (20, 0.9, 0.1).

**Figure 7 sensors-25-03621-f007:**
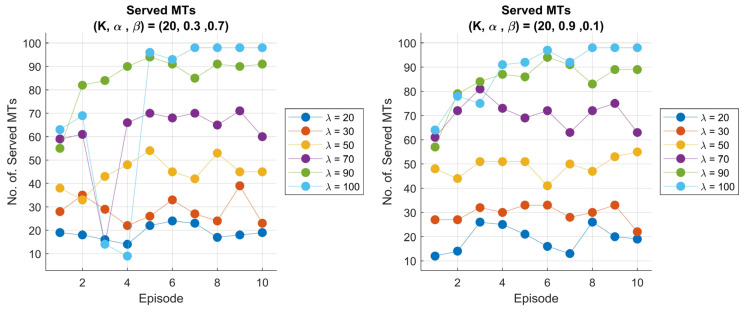
Performance of 6G-RQS with RL integration: total served MTs per episode: for (K, α, β) = (20, 0.3, 0.7) (**left**), and (K, α, β) = (20, 0.1, 0.9) (**right**).

**Figure 8 sensors-25-03621-f008:**
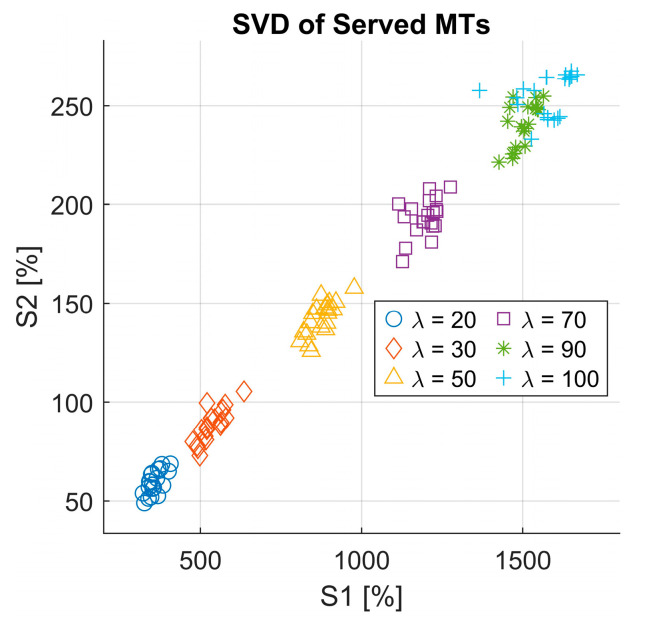
Singular value decomposition of the served MTs.

**Figure 9 sensors-25-03621-f009:**
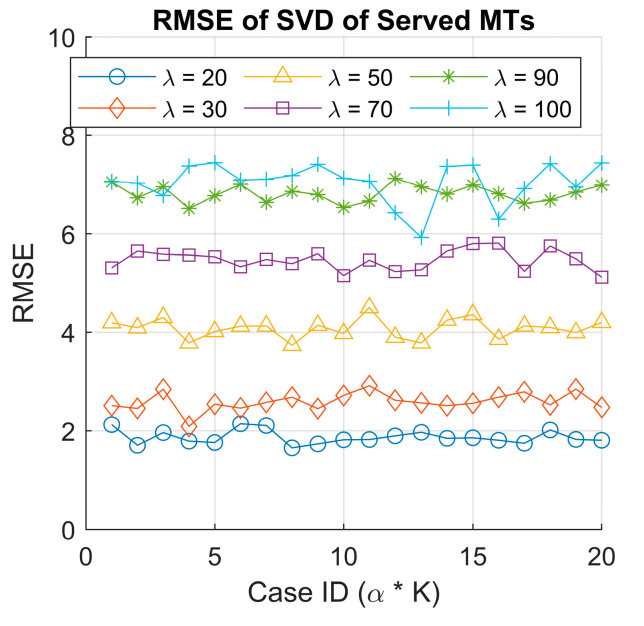
RMSE of the original and recomposed served MT data using singular value decomposition.

**Figure 10 sensors-25-03621-f010:**
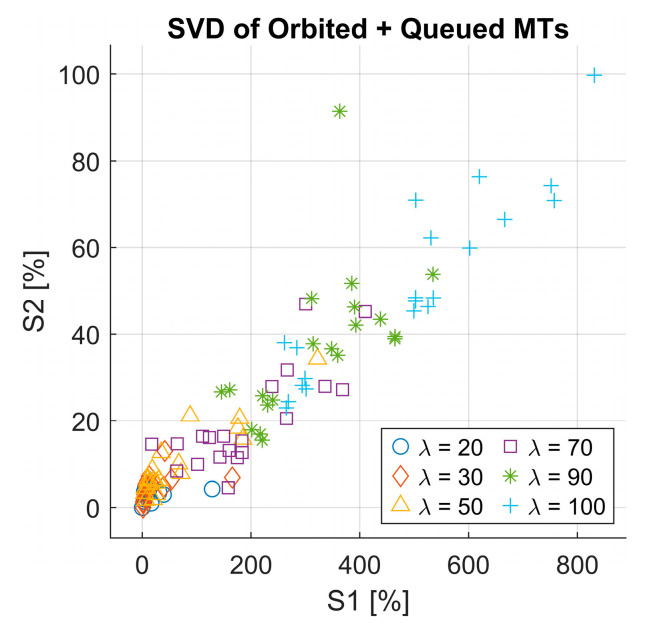
Singular value decomposition of the orbited + queued MTs.

**Figure 11 sensors-25-03621-f011:**
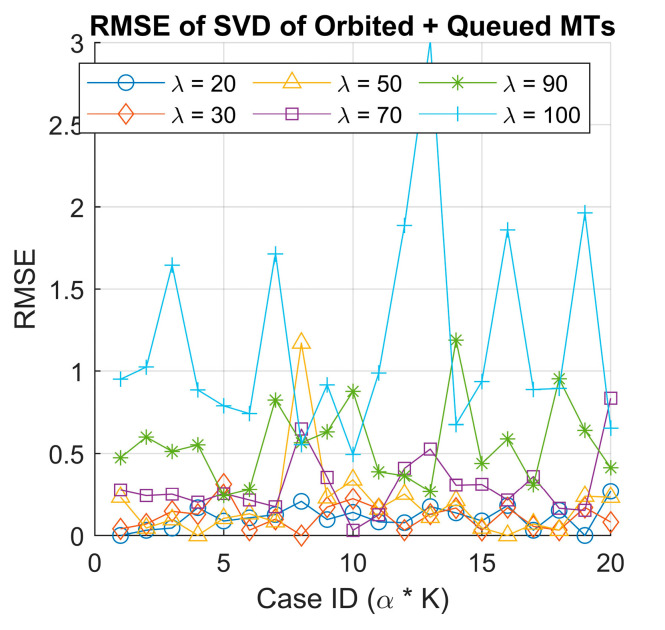
RMSE of the original and recomposed orbited + queued MTs data using singular value decomposition.

**Figure 12 sensors-25-03621-f012:**
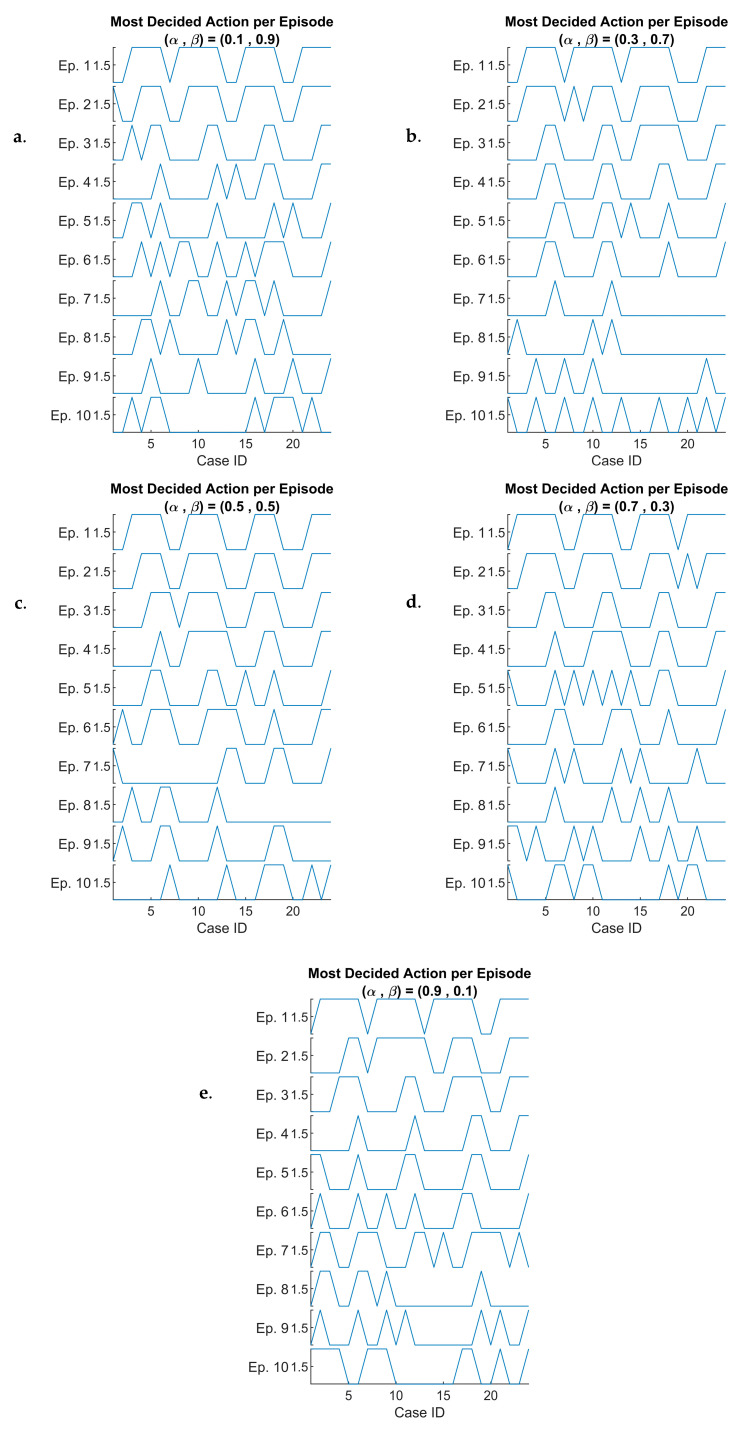
DQN-based most decided actions per episode versus case ID for: (α,β) = (0.1, 0.9) (**a**), (α,β) = (0.3, 0.7) (**b**), (α,β) = (0.5, 0.5) (**c**), (α,β) = (0.7, 0.3) (**d**), and (α,β) = (0.9, 0.1) (**e**).

**Figure 13 sensors-25-03621-f013:**
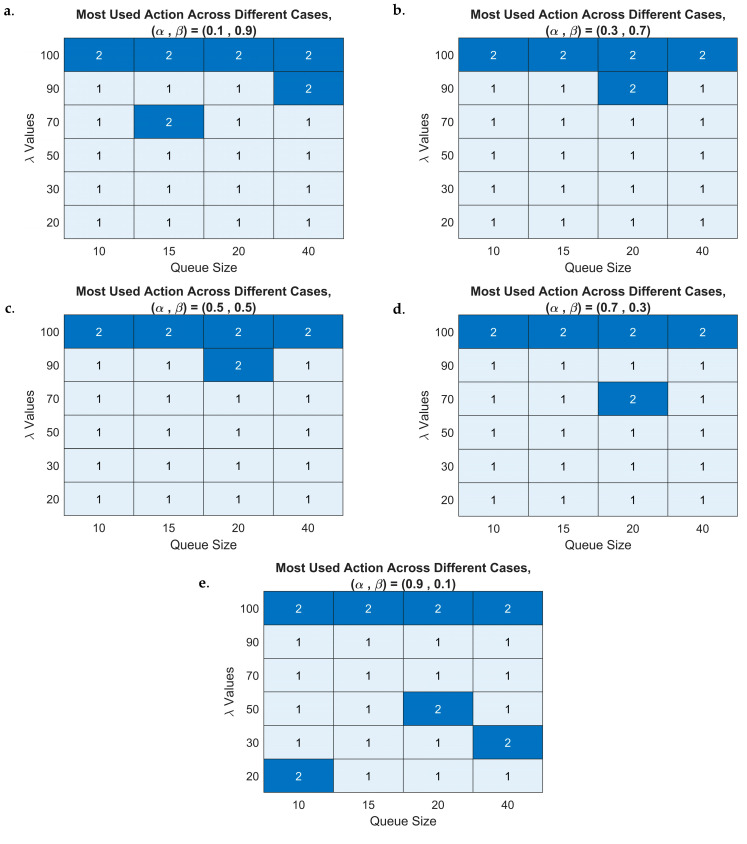
DQN-based most decided actions heatmap for: (α, β) = (0.1, 0.9) (**a**), (α, β) = (0.3, 0.7) (**b**), (α, β) = (0.5, 0.5) (**c**), (α, β) = (0.7, 0.3) (**d**), and (α, β) = (0.9, 0.1) (**e**).

**Figure 14 sensors-25-03621-f014:**
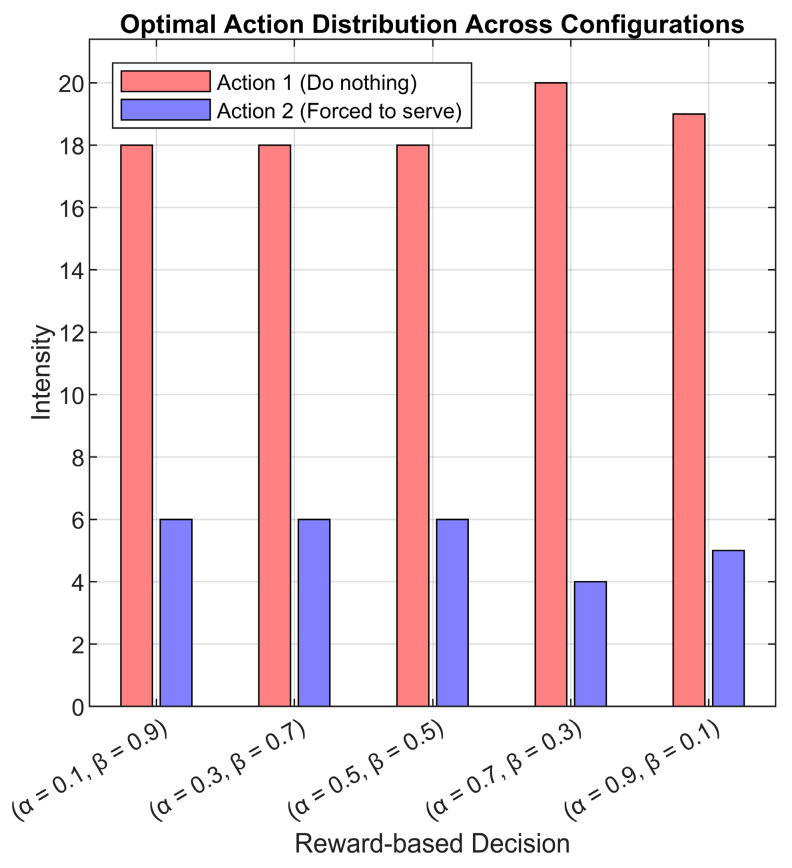
Critic NN learned weights-based most decided actions histogram for all cases.

**Table 1 sensors-25-03621-t001:** Retrial queueing system model used parameters.

Parameter	Value
Arrival rate (λ)	45
Service rate (μ)	40
Retry rate (θ)	20
Queue size (K)	5
Simulation time (T)	1000

**Table 2 sensors-25-03621-t002:** DQN critic network training parameters.

Parameter	Value	Description
Gradient Threshold	1	Limits the magnitude of a gradient to prevent instability.
Learn Rate	$10^{-3}$	Controls the step size in weight updates, ensuring smooth convergence.
Target Smooth Factor	$10^{-1}$	Determines the smoothing factor for updating the target network for stable learning.
Experience Buffer Length [Bytes]	$10^{6}$	Defines the size of the replay buffer used for experience replay, improving sample efficiency.
Discount Factor	0.99	Ensures future rewards are considered.
Mini Batch Size	64	Specified the number of samples used per training step for gradient update.
Number of Episodes	10	Defines the number of training episodes used for learning the policy.

**Table 3 sensors-25-03621-t003:** Used parameters for the simulation.

Parameter	Value
Arrival rate (λ)	[20, 30, 50, 70, 90, 100]
Service rate (μ)	0.5
Retry rate (θ)	1
Queue size (K)	[10, 15, 20, 40]
Scaling factor (α)	[0.1, 0.3, 0.5, 0.7, 0.9]
Weight (β)	1 − α
Simulation time	100

## Data Availability

The data were generated using the algorithms outlined in the article.

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
