# Peer review of "Integrating Reinforcement Learning into M/M/1/K Retry Queueing Models for 6G Applications"

_sensors, 2025, doi:10.3390/s25123621_

Round 1
Reviewer 1 Report
Comments and Suggestions for Authors
This paper proposes a reinforcement learning method based on Deep Q-Networks (DQN) to optimize resource allocation and queue management in the M/M/1/K retrial queueing model for 6G networks. By dynamically adjusting service policies, the study validates through simulations that this approach effectively increases the number of served mobile terminals (MTs) (by 28%) while reducing queue congestion and the backlog of orbited MTs. Singular Value Decomposition (SVD) is employed to analyze latent patterns in system behavior, demonstrating the adaptability of reinforcement learning in the highly dynamic 6G environment. However, there remains room for improvement:
- Questionable applicability of the Poisson process: The paper assumes MT arrivals follow a Poisson process, but the high dynamism of 6G networks (e.g., user mobility, bursty traffic) may cause the actual arrival process to deviate from this assumption. While the authors cite some literature to support this claim (e.g., IEEE 802.11ad), they do not address the impact of THz-specific blockage effects or directional communication on the arrival process.
- Limited state representation in reinforcement learning: The state space includes only queue length and AP status (busy/idle), omitting critical factors such as channel quality, MT priority, and historical delays. This may lead to suboptimal policies, especially in URLLC scenarios.
- Lack of comparative analysis: The study does not compare the proposed method with traditional queue management techniques (e.g., WFQ, RED) or non-RL optimization approaches (e.g., dynamic programming), making it difficult to demonstrate the superiority of RL.
- Practical implementation concerns: The real-time performance of DQN (e.g., whether inference latency meets 6G’s millisecond-level requirements) is not evaluated.
- Incomplete pseudocode: Key logic for RL-queue interaction (e.g., specific rules for state updates) is not fully detailed in Algorithms 1 and 2.
- Unaddressed anomalies: Performance fluctuations (e.g., the drop for α=0.3 in Figure 7) are not thoroughly analyzed.
- Redundancy and coherence issues: Some sentences are repetitive, and transitions could be improved to enhance readability.
8. The following work related to Reinforcement Learning for 6G Applications is missed,
“Distributed Deep Reinforcement Learning Based Gradient Quantization for Federated Learning Enabled Vehicle Edge Computing,” IEEE Internet of Things Journal, Vol. 12, No. 5, Mar. 2025, pp. 4899-4913.
Comments on the Quality of English LanguagePlease refer to comments and suggests for authros.
Author Response
Please see the attachment containing the answers.
Manuscript number: sensors-3646855
Manuscript title: Integrating Reinforcement Learning into M/M/1/K Retry Queueing Models for 6G Applications
Date: 31-05-2025.
Response to Reviewers:
Dear Editor/Reviewers,
We would like to express our sincere gratitude for the insightful feedback provided by you and the reviewers. We have thoroughly considered each comment and implemented the necessary revisions to improve our manuscript. Below, we present our responses to each comment, followed by the original feedback from the reviewers.
Reviewer #1 comments:
Comment 1: Questionable applicability of the Poisson process: The paper assumes MT arrivals follow a Poisson process, but the high dynamism of 6G networks (e.g., user mobility, bursty traffic) may cause the actual arrival process to deviate from this assumption. While the authors cite some literature to support this claim (e.g., IEEE 802.11ad), they do not address the impact of THz-specific blockage effects or directional communication on the arrival process.
Response: We thank the reviewer for this insightful comment. In response, we have revised the manuscript to acknowledge the limitations of the Poisson process assumption in the context of highly dynamic 6G THz networks, particularly under user mobility, bursty traffic, and blockage scenarios. The added clarification ensures the assumptions are well-grounded while remaining open to future extensions. You can find the added part in ‘section 3.1’.
Comment 2: Limited state representation in reinforcement learning: The state space includes only queue length and AP status (busy/idle), omitting critical factors such as channel quality, MT priority, and historical delays. This may lead to suboptimal policies, especially in URLLC scenarios.
Response: We appreciate the reviewer’s insightful observation regarding the limited state representation in our reinforcement learning model. Further clarification regarding the choice of a simplified state space and its implications has been added to the manuscript in ‘Section 4.2.’
Comment 3: Lack of comparative analysis: The study does not compare the proposed method with traditional queue management techniques (e.g., WFQ, RED) or non-RL optimization approaches (e.g., dynamic programming), making it difficult to demonstrate the superiority of RL.
Response: We thank the reviewer for this valuable comment. As presented in Section 3, we first evaluate the baseline performance of the RQS model without DQN-RL, highlighting its limitations under dynamic traffic and service conditions. Then, in Section 4, we demonstrate the enhanced performance after integrating DQN-based reinforcement learning. To further address the comment, a direct comparative discussion between the two models (with and without RL) has now been added to Section 5 of the manuscript.
Comment 4: Practical implementation concerns: The real-time performance of DQN (e.g., whether inference latency meets 6G’s millisecond-level requirements) is not evaluated.
Response:
We appreciate the reviewer’s observation. A clarification regarding the practical inference latency of the DQN model has been added to Section 4.2, within the discussion of Figure 5. This addition explains that the inference phase, which involves only a forward pass through the trained neural network, operates within sub-millisecond latency on standard edge AI hardware, ensuring compatibility with the strict latency constraints of 6G systems.
Comment 5: Incomplete pseudocode: Key logic for RL-queue interaction (e.g., specific rules for state updates) is not fully detailed in Algorithms 1 and 2.
Response:
We appreciate the reviewer’s careful reading and valuable feedback regarding the pseudocode presented in Algorithms 1 and 2. We would like to clarify that Algorithm 1 represents the baseline Retrial Queueing System (RQS) model without reinforcement learning integration; consequently, it does not involve explicit state updates related to RL, and thus the state update logic is inherently not applicable in this context. The algorithm fully details the queueing and retrial dynamics as intended.
Regarding Algorithm 2, which incorporates the reinforcement learning component, we acknowledge that state update logic is essential and have accordingly included the necessary rules and procedures for state updates. These have been explicitly detailed in Algorithm 2 to ensure completeness and clarity.
We trust this addresses the concern satisfactorily and we remain open to further suggestions to improve clarity.
Comment 6: Unaddressed anomalies: Performance fluctuations (e.g., the drop for α=0.3 in Figure 7) are not thoroughly analyzed.
Response:
We thank the reviewer for pointing this out. The performance fluctuations for α = 0.3 have been explicitly discussed in Section 4.2 during the analysis of Figure 7. We explain the drop in the number of served MTs and the increased fluctuations as consequences of the low reward scaling factor α, which leads to suboptimal decision-making during early learning stages. The explanation contrasts this with the behaviour observed for α = 0.9 and elaborates on the implications for system stability and agent motivation.
Comment 7: Redundancy and coherence issues: Some sentences are repetitive, and transitions could be improved to enhance readability.
Response:
We appreciate the reviewer’s observation. We have revised the manuscript to remove redundancies and improve the coherence and transitions between paragraphs for better readability.
Comment 8: The following work related to Reinforcement Learning for 6G Applications is missed: “Distributed Deep Reinforcement Learning Based Gradient Quantization for Federated Learning Enabled Vehicle Edge Computing,” IEEE Internet of Things Journal, Vol. 12, No. 5, Mar. 2025, pp. 4899-4913.”
Response:
We thank the reviewer for the suggestion. The referenced work has now been added and cited appropriately in the Related Work Section 2.2

Reviewer 2 Report
Comments and Suggestions for Authors
In this paper the authors apply finite capacity retrial queueing models of the type M/M/1 to 6G applications.
The paper considers an interesting application useful in practice but poorly written as pointed below.
The authors need to revise the paper taking into account the following comments/concerns/suggestions.
Without revision the paper should be rejected.
- RL is used but not defined even though the tile has the meaning of RL. But the body of the text should have this.
- The concept “intelligent queueing…” is used but need to define what this is before using this terminology.
- Key references (to terminology and the results) need to be added where and when referenced. For example, see Section 3 wherein Kendall’s notation and queueing models are referred but no references given. This is very important for readers (who are using 6G applications) who may not be aware of such queues. Also in Section 4, key concepts like reinforcement learning, Q-network, Q-learning, are all used but no references given.
- Why Poisson process for the input when so many other useful and versatile point processes are available? Justify this.
- Why exponential services when so many other useful and versatile distributions are available for services? Justify this.
- I would suggest that the authors have a separate section to introduce key concepts (with references) to make the paper readable and also for practitioners to appreciate what queueing models can do for them.
- Notations are used without defining them. See e.g. Δ. Only after reading the paragraph following equation (13) one sees what this is. Move this before defining (13) and no need to define the queue and the served.
- Avoid using “.” in (14) (and elsewhere) and write as one whole quantity like it is seen in matrix algebra papers and books.
- While the numerical examples presented are very illustrative my concerns are with regard to the choice for K. Shouldn’t the authors consider even for large values running in 1000s? What are the buffer sizes in real-life applications? Justify with some practical situations?
- Did the authors validate some of their conclusions with real-life cases? If so, mention. If not, why not?
- Time (100 units mentioned) for simulation seems to be less. What is the justification for this choice? Basically an average of only 2000 to 10000 arrivals are considered during this simulation. Is that what is seen in real life situations? Justify.
Author Response
Please see the attachment containing the answers.
Manuscript number: sensors-3646855
Manuscript title: Integrating Reinforcement Learning into M/M/1/K Retry Queueing Models for 6G Applications
Date: 31-05-2025.
Response to Reviewers:
Dear Editor/Reviewers,
We would like to express our sincere gratitude for the insightful feedback provided by you and the reviewers. We have thoroughly considered each comment and implemented the necessary revisions to improve our manuscript. Below, we present our responses to each comment, followed by the original feedback from the reviewers.
Reviewer #2 comments:
Comment 1: RL is used but not defined, even though the title has the meaning of RL. But the body of the text should have this.
Response: Thank you for your valuable feedback. We acknowledge the oversight in not explicitly defining Reinforcement Learning (RL) in the main text. We have now added a clear and concise definition of RL in the Introduction section to ensure clarity for all readers.
Comment 2: The concept “intelligent queueing…” is used but need to define what this is before using this terminology.
Response: Thank you for your insightful comment. We have addressed the issue by including a clear definition of "intelligent queueing" in Section 1.1.
Comment 3: Key references (to terminology and the results) need to be added where and when referenced. For example, see Section 3 wherein Kendall’s notation and queueing models are referred but no references given. This is very important for readers (who are using 6G applications) who may not be aware of such queues. Also in Section 4, key concepts like reinforcement learning, Q-network, Q-learning, are all used but no references given.
Response: We thank the reviewer for the comment. Appropriate references have now been added in Sections 3 and 4 to support the mentioned concepts and terminologies.
Comment 4: Why Poisson process for the input when so many other useful and versatile point processes are available? Justify this.
Response:
We thank the reviewer for this valuable observation. In the revised manuscript, we have expanded the justification for using the Poisson process in the arrival modeling context in Section 3.1.
Comment 5: Why exponential services when so many other useful and versatile distributions are available for services? Justify this.
Response:
We thank the reviewer for the comment. The justification for using exponential service times has been added in Section 3.1 of the revised manuscript.
Comment 6: I would suggest that the authors have a separate section to introduce key concepts (with references) to make the paper readable and also for practitioners to appreciate what queueing models can do for them.
Response:
We thank the reviewer for the valuable suggestion to include a dedicated section introducing the key concepts of queueing theory. In response, we have added a new subsection titled 2.4 Fundamentals of Queueing Theory and Its Applications in Wireless Networks within the Related Work section. This subsection provides a concise yet comprehensive overview of queueing models, their relevance, and practical applications in wireless communication systems, supported by appropriate references. We believe this addition enhances the readability of the paper and helps practitioners better appreciate the role and benefits of queueing theory in the context of 6G network applications.
Comment 7: Notations are used without defining them. See e.g. Δ. Only after reading the paragraph following equation (13) one sees what this is. Move this before defining (13) and no need to define the queue and the served.
Response:
We appreciate the reviewer’s careful reading and helpful feedback regarding the notation usage. We have moved the definition of Δ to appear before equation (13) as suggested and removed the definitions of "queue" and "served".
Comment 8: Avoid using “.” in (14) (and elsewhere) and write as one whole quantity like it is seen in matrix algebra papers and books.
Response:
We thank the reviewer for the valuable suggestion. We have revised equation (14) and other relevant expressions by removing the “.” Notation.
Comment 9: While the numerical examples presented are very illustrative my concerns are with regard to the choice for K. Shouldn’t the authors consider even for large values running in 1000s? What are the buffer sizes in real-life applications? Justify with some practical situations?
Response:
We appreciate the reviewer’s insightful observation regarding the selection of the queue size K. In response, we have added a detailed explanation in Section 4.3, following Table 3, clarifying that K is varied between 10 and 40 to reflect realistic edge scenarios such as URLLC and mission-critical IoT applications, where buffers are intentionally limited to meet strict latency requirements. This range aligns with practical deployments and 3GPP specifications emphasizing minimal buffering at the network edge, thereby strengthening the practical relevance and justification of our simulation parameters.
Comment 10: Did the authors validate some of their conclusions with real-life cases? If so, mention. If not, why not?
Response:
We thank the reviewer for this important question. We acknowledge the importance of validating findings with real-life data. However, no real-life datasets are currently available for the type of next-generation mobile technologies (i.e., 6G) targeted in this study, as 6G networks are still under active development and are expected to become commercially accessible towards the end of this decade. Therefore, our conclusions are based on simulation-based evaluations and theoretical modeling, which is a standard and accepted approach in early-stage research on future network architectures. Nevertheless, we have structured our framework to be adaptable, and we plan to validate it with empirical data as soon as relevant deployments and measurement campaigns become available.
Comment 11: Time (100 units mentioned) for simulation seems to be less. What is the justification for this choice? Basically an average of only 2000 to 10000 arrivals are considered during this simulation. Is that what is seen in real life situations? Justify.
Response:
We thank the reviewer for raising this important point. In the manuscript, we have added a detailed explanation regarding the choice of the simulation time of 100 time units in Section 4.3 in table 3 explanation. This duration balances computational efficiency and the need to capture meaningful queueing dynamics for effective reinforcement learning convergence.

Round 2
Reviewer 1 Report
Comments and Suggestions for Authors
This paper can be accepted.
Author Response
Manuscript number: sensors-3646855
Manuscript title: Integrating Reinforcement Learning into M/M/1/K Retry Queueing Models for 6G Applications
Date: 04-06-2025.
Response to Reviewers:
Dear Editor/Reviewers,
We sincerely thank you for the positive feedback and the recommendation for acceptance. We appreciate the time and effort dedicated to reviewing our work and are grateful for the constructive suggestions that helped us improve the manuscript.
Reviewer #1 comments:
Comment 1: This paper can be accepted.
Response: We sincerely thank the reviewer for their positive evaluation and recommendation for acceptance. We appreciate the time and effort dedicated to reviewing our work.

Reviewer 2 Report
Comments and Suggestions for Authors
The revision has incorporated the suggestions. The paper can be accepted for publication. Only comment (which I forgot to mention) is to add future work.
Author Response
Manuscript number: sensors-3646855
Manuscript title: Integrating Reinforcement Learning into M/M/1/K Retry Queueing Models for 6G Applications
Date: 04-06-2025.
Response to Reviewers:
Dear Editor/Reviewers,
We sincerely thank you for the positive feedback and the recommendation for acceptance. We appreciate the time and effort dedicated to reviewing our work and are grateful for the constructive suggestions that helped us improve the manuscript.
Reviewer #2 comments:
Comment 1: The revision has incorporated the suggestions. The paper can be accepted for publication. Only comment (which I forgot to mention) is to add future work.
Response: We thank the reviewer for the positive feedback and final comment. As noted, future work is already discussed in detail in “Section 5 Discussion of the RL DQN Simulation Findings” and summarized in the final paragraph of “Section 6 Conclusion”. We have adjusted the relevant paragraphs to ensure that the future work directions are clearly stated and easily visible to readers.
